# Scalable trust-region method for deep reinforcement learning using Kronecker-factored approximation

**Yuhuai Wu**[*]
University of Toronto
Vector Institute
ywu@cs.toronto.edu

**Elman Mansimov**[*]
New York University
mansimov@cs.nyu.edu

**Shun Liao**
University of Toronto
Vector Institute
sliao3@cs.toronto.edu

**Roger Grosse**
University of Toronto
Vector Institute
rgrosse@cs.toronto.edu

**Jimmy Ba**
University of Toronto
Vector Institute
jimmy@psi.utoronto.ca

## Abstract

In this work, we propose to apply trust region optimization to deep reinforcement learning using a recently proposed Kronecker-factored approximation to the curvature. We extend the framework of natural policy gradient and propose to optimize both the actor and the critic using Kronecker-factored approximate curvature (K-FAC) with trust region; hence we call our method Actor Critic using Kronecker-Factored Trust Region (ACKTR). To the best of our knowledge, this is the first scalable trust region natural gradient method for actor-critic methods. It is also the method that learns non-trivial tasks in continuous control as well as discrete control policies directly from raw pixel inputs. We tested our approach across discrete domains in Atari games as well as continuous domains in the Mu-JoCo environment. With the proposed methods, we are able to achieve higher rewards and a 2- to 3-fold improvement in sample efficiency on average, compared to previous state-of-the-art on-policy actor-critic methods. Code is available at https://github.com/openai/baselines.

## 1 Introduction

Agents using deep reinforcement learning (deep RL) methods have shown tremendous success in learning complex behaviour skills and solving challenging control tasks in high-dimensional raw sensory state-space [24, 17, 12]. Deep RL methods make use of deep neural networks to represent control policies. Despite the impressive results, these neural networks are still trained using simple variants of stochastic gradient descent (SGD). SGD and related first-order methods explore weight space inefficiently. It often takes days for the current deep RL methods to master various continuous and discrete control tasks. Previously, a distributed approach was proposed [17] to reduce training time by executing multiple agents to interact with the environment simultaneously, but this leads to rapidly diminishing returns of sample efficiency as the degree of parallelism increases.

Sample efficiency is a dominant concern in RL; robotic interaction with the real world is typically scarcer than computation time, and even in simulated environments the cost of simulation often dominates that of the algorithm itself. One way to effectively reduce the sample size is to use more advanced optimization techniques for gradient updates. Natural policy gradient [10] uses the technique of natural gradient descent [1] to perform gradient updates. Natural gradient methods

---

[*]Equal contribution.

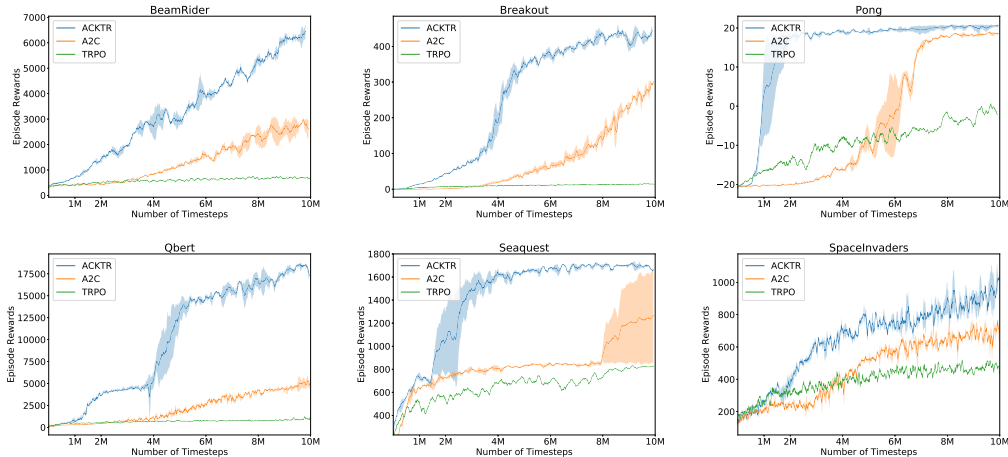

Figure 1: Performance comparisons on six standard Atari games trained for 10 million timesteps (1 timestep equals 4 frames). The shaded region denotes the standard deviation over 2 random seeds.

follow the steepest descent direction that uses the Fisher metric as the underlying metric, a metric that is based not on the choice of coordinates but rather on the manifold (i.e., the surface).

However, the exact computation of the natural gradient is intractable because it requires inverting the Fisher information matrix. Trust-region policy optimization (TRPO) [21] avoids explicitly storing and inverting the Fisher matrix by using Fisher-vector products [20]. However, it typically requires many steps of conjugate gradient to obtain a single parameter update, and accurately estimating the curvature requires a large number of samples in each batch; hence TRPO is impractical for large models and suffers from sample inefficiency.

Kronecker-factored approximated curvature (K-FAC) [15, 6] is a scalable approximation to natural gradient. It has been shown to speed up training of various state-of-the-art large-scale neural networks [2] in supervised learning by using larger mini-batches. Unlike TRPO, each update is comparable in cost to an SGD update, and it keeps a running average of curvature information, allowing it to use small batches. This suggests that applying K-FAC to policy optimization could improve the sample efficiency of the current deep RL methods.

In this paper, we introduce the actor-critic using Kronecker-factored trust region (ACKTR; pronounced "actor") method, a scalable trust-region optimization algorithm for actor-critic methods. The proposed algorithm uses a Kronecker-factored approximation to natural policy gradient that allows the covariance matrix of the gradient to be inverted efficiently. To best of our knowledge, we are also the first to extend the natural policy gradient algorithm to optimize value functions via Gauss-Newton approximation. In practice, the per-update computation cost of ACKTR is only $10\%$ to $25\%$ higher than SGD-based methods. Empirically, we show that ACKTR substantially improves both sample efficiency and the final performance of the agent in the Atari environments [4] and the MuJoCo [26] tasks compared to the state-of-the-art on-policy actor-critic method A2C [17] and the famous trust region optimizer TRPO [21].

We make our source code available online at `https://github.com/openai/baselines`.

## 2 Background

### 2.1 Reinforcement learning and actor-critic methods

We consider an agent interacting with an infinite-horizon, discounted Markov Decision Process $(\mathcal{X}, \mathcal{A}, \gamma, P, r)$. At time $t$, the agent chooses an action $a_t \in \mathcal{A}$ according to its policy $\pi_\theta(a|s_t)$ given its current state $s_t \in \mathcal{X}$. The environment in turn produces a reward $r(s_t, a_t)$ and transitions to the next state $s_{t+1}$ according to the transition probability $P(s_{t+1}|s_t, a_t)$. The goal of the agent is to maximize the expected $\gamma$-discounted cumulative return $\mathcal{J}(\theta) = \mathbb{E}_\pi[R_t] = \mathbb{E}_\pi[\sum_{i \geq 0} \gamma^i r(s_{t+i}, a_{t+i})]$ with respect to the policy parameters $\theta$. Policy gradient methods [28, 25] directly parameterize a policy $\pi_\theta(a|s_t)$ and update parameter $\theta$ so as to maximize the objective $\mathcal{J}(\theta)$. In its general form,

the policy gradient is defined as [22],

$$\nabla_\theta \mathcal{J}(\theta) = \mathbb{E}_\pi [\sum_{t=0}^{\infty} \Psi^t \nabla_\theta \log \pi_\theta(a_t|s_t)],$$

where $\Psi^t$ is often chosen to be the advantage function $A^\pi(s_t, a_t)$, which provides a relative measure of value of each action $a_t$ at a given state $s_t$. There is an active line of research [22] on designing an advantage function that provides both low-variance and low-bias gradient estimates. As this is not the focus of our work, we simply follow the asynchronous advantage actor critic (A3C) method [17] and define the advantage function as the $k$-step returns with function approximation,

$$A^\pi(s_t, a_t) = \sum_{i=0}^{k-1} \gamma^i r(s_{t+i}, a_{t+i}) + \gamma^k V_\phi^\pi(s_{t+k}) - V_\phi^\pi(s_t),$$

where $V_\phi^\pi(s_t)$ is the value network, which provides an estimate of the expected sum of rewards from the given state following policy $\pi$, $V_\phi^\pi(s_t) = \mathbb{E}_\pi[R_t]$. To train the parameters of the value network, we again follow [17] by performing temporal difference updates, so as to minimize the squared difference between the bootstrapped $k$-step returns $\hat{R}_t$ and the prediction value $\frac{1}{2}||\hat{R}_t - V_\phi^\pi(s_t)||^2$.

## 2.2 Natural gradient using Kronecker-factored approximation

To minimize a nonconvex function $\mathcal{J}(\theta)$, the method of steepest descent calculates the update $\Delta\theta$ that minimizes $\mathcal{J}(\theta + \Delta\theta)$, subject to the constraint that $||\Delta\theta||_B < 1$, where $||\cdot||_B$ is the norm defined by $||x||_B = (x^T B x)^{\frac{1}{2}}$, and $B$ is a positive semidefinite matrix. The solution to the constraint optimization problem has the form $\Delta\theta \propto -B^{-1}\nabla_\theta \mathcal{J}$, where $\nabla_\theta \mathcal{J}$ is the standard gradient. When the norm is Euclidean, i.e., $B = I$, this becomes the commonly used method of gradient descent. However, the Euclidean norm of the change depends on the parameterization $\theta$. This is not favorable because the parameterization of the model is an arbitrary choice, and it should not affect the optimization trajectory. The method of natural gradient constructs the norm using the Fisher information matrix $F$, a local quadratic approximation to the KL divergence. This norm is independent of the model parameterization $\theta$ on the class of probability distributions, providing a more stable and effective update. However, since modern neural networks may contain millions of parameters, computing and storing the exact Fisher matrix and its inverse is impractical, so we have to resort to approximations.

A recently proposed technique called Kronecker-factored approximate curvature (K-FAC) [15] uses a Kronecker-factored approximation to the Fisher matrix to perform efficient approximate natural gradient updates. We let $p(y|x)$ denote the output distribution of a neural network, and $L = \log p(y|x)$ denote the log-likelihood. Let $W \in \mathbb{R}^{C_{out} \times C_{in}}$ be the weight matrix in the $\ell^{\text{th}}$ layer, where $C_{out}$ and $C_{in}$ are the number of output/input neurons of the layer. Denote the input activation vector to the layer as $a \in \mathbb{R}^{C_{in}}$, and the pre-activation vector for the next layer as $s = Wa$. Note that the weight gradient is given by $\nabla_W L = (\nabla_s L)a^\intercal$. K-FAC utilizes this fact and further approximates the block $F_\ell$ corresponding to layer $\ell$ as $\hat{F}_\ell$,

$$F_\ell = \mathbb{E}[\text{vec}\{\nabla_W L\}\text{vec}\{\nabla_W L\}^\intercal] = \mathbb{E}[aa^\intercal \otimes \nabla_s L(\nabla_s L)^\intercal]$$

$$\approx \mathbb{E}[aa^\intercal] \otimes \mathbb{E}[\nabla_s L(\nabla_s L)^\intercal] := A \otimes S := \hat{F}_\ell,$$

where $A$ denotes $\mathbb{E}[aa^\intercal]$ and $S$ denotes $\mathbb{E}[\nabla_s L(\nabla_s L)^\intercal]$. This approximation can be interpreted as making the assumption that the second-order statistics of the activations and the backpropagated derivatives are uncorrelated. With this approximation, the natural gradient update can be efficiently computed by exploiting the basic identities $(P \otimes Q)^{-1} = P^{-1} \otimes Q^{-1}$ and $(P \otimes Q)\text{vec}(T) = PTQ^\intercal$:

$$\text{vec}(\Delta W) = \hat{F}_\ell^{-1} \text{vec}\{\nabla_W \mathcal{J}\} = \text{vec}\left(A^{-1} \nabla_W \mathcal{J} S^{-1}\right).$$

From the above equation we see that the K-FAC approximate natural gradient update only requires computations on matrices comparable in size to $W$. Grosse and Martens [6] have recently extended the K-FAC algorithm to handle convolutional networks. Ba et al. [2] later developed a distributed version of the method where most of the overhead is mitigated through asynchronous computation. Distributed K-FAC achieved 2- to 3-times speed-ups in training large modern classification convolutional networks.

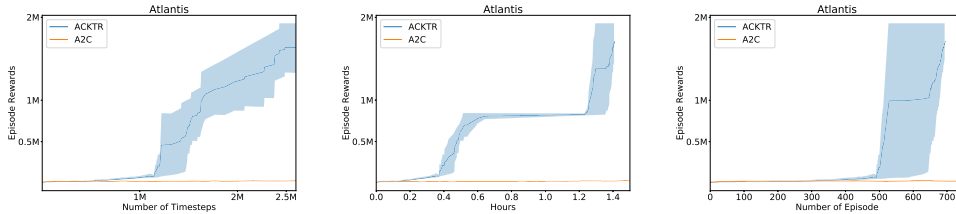

Figure 2: In the Atari game of Atlantis, our agent (ACKTR) quickly learns to obtain rewards of 2 million in 1.3 hours, 600 episodes of games, 2.5 million timesteps. The same result is achieved by advantage actor critic (A2C) in 10 hours, 6000 episodes, 25 million timesteps. ACKTR is 10 times more sample efficient than A2C on this game.

## 3  Methods

### 3.1  Natural gradient in actor-critic

Natural gradient was proposed to apply to the policy gradient method more than a decade ago by Kakade [10]. But there still doesn't exist a scalable, sample-efficient, and general-purpose instantiation of the natural policy gradient. In this section, we introduce the first scalable and sample-efficient natural gradient algorithm for actor-critic methods: the actor-critic using Kronecker-factored trust region (ACKTR) method. We use Kronecker-factored approximation to compute the natural gradient update, and apply the natural gradient update to both the actor and the critic.

To define the Fisher metric for reinforcement learning objectives, one natural choice is to use the policy function which defines a distribution over the action given the current state, and take the expectation over the trajectory distribution:

$$F = \mathbb{E}_{p(\tau)}[\nabla_\theta \log \pi(a_t|s_t)(\nabla_\theta \log \pi(a_t|s_t))^\intercal],$$

where $p(\tau)$ is the distribution of trajectories, given by $p(s_0) \prod_{t=0}^{T} \pi(a_t|s_t)p(s_{t+1}|s_t, a_t)$. In practice, one approximates the intractable expectation over trajectories collected during training.

We now describe one way to apply natural gradient to optimize the critic. Learning the critic can be thought of as a least-squares function approximation problem, albeit one with a moving target. In the setting of least-squares function approximation, the second-order algorithm of choice is commonly Gauss-Newton, which approximates the curvature as the Gauss-Newton matrix $G := \mathbb{E}[J^T J]$, where $J$ is the Jacobian of the mapping from parameters to outputs [18]. The Gauss-Newton matrix is equivalent to the Fisher matrix for a Gaussian observation model [14]; this equivalence allows us to apply K-FAC to the critic as well. Specifically, we assume the output of the critic $v$ is defined to be a Gaussian distribution $p(v|s_t) \sim \mathcal{N}(v; V(s_t), \sigma^2)$. The Fisher matrix for the critic is defined with respect to this Gaussian output distribution. In practice, we can simply set $\sigma$ to 1, which is equivalent to the vanilla Gauss-Newton method.

If the actor and critic are disjoint, one can separately apply K-FAC updates to each using the metrics defined above. But to avoid instability in training, it is often beneficial to use an architecture where the two networks share lower-layer representations but have distinct output layers [17, 27]. In this case, we can define the joint distribution of the policy and the value distribution by assuming independence of the two output distributions, i.e., $p(a, v|s) = \pi(a|s)p(v|s)$, and construct the Fisher metric with respect to $p(a, v|s)$, which is no different than the standard K-FAC except that we need to sample the networks' outputs independently. We can then apply K-FAC to approximate the Fisher matrix $\mathbb{E}_{p(\tau)}[\nabla \log p(a, v|s)\nabla \log p(a, v|s)^T]$ to perform updates simultaneously.

In addition, we use regular damping for regularization. We also follow [2] and perform the asynchronous computation of second-order statistics and inverses required by the Kronecker approximation to reduce computation time.

## 3.2 Step-size Selection and trust-region optimization

Traditionally, natural gradient is performed with SGD-like updates, $\theta \leftarrow \theta - \eta F^{-1}\nabla_\theta L$. But in the context of deep RL, Schulman et al. [21] observed that such an update rule can result in large updates to the policy, causing the algorithm to prematurely converge to a near-deterministic policy. They advocate instead using a trust region approach, whereby the update is scaled down to modify the policy distribution (in terms of KL divergence) by at most a specified amount. Therefore, we adopt the trust region formulation of K-FAC introduced by [2], choosing the effective step size $\eta$ to be $\min(\eta_{\max}, \sqrt{\frac{2\delta}{\Delta\theta^\top \hat{F}\Delta\theta}})$, where the learning rate $\eta_{\max}$ and trust region radius $\delta$ are hyperparameters. If the actor and the critic are disjoint, then we need to tune a different set of $\eta_{\max}$ and $\delta$ separately for both. The variance parameter for the critic output distribution can be absorbed into the learning rate parameter for vanilla Gauss-Newton. On the other hand, if they share representations, we need to tune one set of $\eta_{\max}$, $\delta$, and also the weighting parameter of the training loss of the critic, with respect to that of the actor.

# 4 Related work

Natural gradient [1] was first applied to policy gradient methods by Kakade [10]. Bagnell and Schneider [3] further proved that the metric defined in [10] is a covariant metric induced by the path-distribution manifold. Peters and Schaal [19] then applied natural gradient to the actor-critic algorithm. They proposed performing natural policy gradient for the actor's update and using a least-squares temporal difference (LSTD) method for the critic's update. However, there are great computational challenges when applying natural gradient methods, mainly associated with efficiently storing the Fisher matrix as well as computing its inverse. For tractability, previous work restricted the method to using the compatible function approximator (a linear function approximator). To avoid the computational burden, Trust Region Policy Optimization (TRPO) [21] approximately solves the linear system using conjugate gradient with fast Fisher matrix-vector products, similar to the work of Martens [13]. This approach has two main shortcomings. First, it requires repeated computation of Fisher vector products, preventing it from scaling to the larger architectures typically used in experiments on learning from image observations in Atari and MuJoCo. Second, it requires a large batch of rollouts in order to accurately estimate curvature. K-FAC avoids both issues by using tractable Fisher matrix approximations and by keeping a running average of curvature statistics during training. Although TRPO shows better per-iteration progress than policy gradient methods trained with first-order optimizers such as Adam [11], it is generally less sample efficient.

Several methods were proposed to improve the computational efficiency of TRPO. To avoid repeated computation of Fisher-vector products, Wang et al. [27] solve the constrained optimization problem with a linear approximation of KL divergence between a running average of the policy network and the current policy network. Instead of the hard constraint imposed by the trust region optimizer, Heess et al. [8] and Schulman et al. [23] added a KL cost to the objective function as a soft constraint. Both papers show some improvement over vanilla policy gradient on continuous and discrete control tasks in terms of sample efficiency.

There are other recently introduced actor-critic models that improve sample efficiency by introducing experience replay [27], [7] or auxiliary objectives [9]. These approaches are orthogonal to our work, and could potentially be combined with ACKTR to further enhance sample efficiency.

# 5 Experiments

We conducted a series of experiments to investigate the following questions: (1) How does ACKTR compare with the state-of-the-art on-policy method and common second-order optimizer baseline in terms of sample efficiency and computational efficiency? (2) What makes a better norm for optimization of the critic? (3) How does the performance of ACKTR scale with batch size compared to the first-order method?

We evaluated our proposed method, ACKTR, on two standard benchmark platforms. We first evaluated it on the discrete control tasks defined in OpenAI Gym [5], simulated by Arcade Learning Environment [4], a simulator for Atari 2600 games which is commonly used as a deep reinforcement learning benchmark for discrete control. We then evaluated it on a variety of continuous control

| Domain | Human level | ACKTR | | A2C | | TRPO (10 M) | |
|---|---|---|---|---|---|---|---|
| | | Rewards | Episode | Rewards | Episode | Rewards | Episode |
| Beamrider | 5775.0 | **13581.4** | **3279** | 8148.1 | 8930 | 670.0 | N/A |
| Breakout | 31.8 | **735.7** | **4094** | 581.6 | 14464 | 14.7 | N/A |
| Pong | 9.3 | **20.9** | **904** | 19.9 | 4768 | -1.2 | N/A |
| Q-bert | 13455.0 | **21500.3** | **6422** | 15967.4 | 19168 | 971.8 | N/A |
| Seaquest | **20182.0** | 1776.0 | N/A | 1754.0 | N/A | 810.4 | N/A |
| Space Invaders | 1652.0 | **19723.0** | **14696** | 1757.2 | N/A | 465.1 | N/A |

Table 1: ACKTR and A2C results showing the last 100 average episode rewards attained after 50 million timesteps, and TRPO results after 10 million timesteps. The table also shows the episode $N$, where $N$ denotes the first episode for which the mean episode reward over the $N^{th}$ game to the $(N + 100)^{th}$ game crosses the human performance level [16], averaged over 2 random seeds.

benchmark tasks defined in OpenAI Gym [5], simulated by the MuJoCo [26] physics engine. Our baselines are (a) a synchronous and batched version of the asynchronous advantage actor critic model (A3C) [17], henceforth called A2C (advantage actor critic), and (b) TRPO [21]. ACKTR and the baselines use the same model architecture except for the TRPO baseline on Atari games, with which we are limited to using a smaller architecture because of the computing burden of running a conjugate gradient inner-loop. See the appendix for other experiment details.

## 5.1 Discrete control

We first present results on the standard six Atari 2600 games to measure the performance improvement obtained by ACKTR. The results on the six Atari games trained for 10 million timesteps are shown in Figure 1, with comparison to A2C and TRPO[2]. ACKTR significantly outperformed A2C in terms of sample efficiency (i.e., speed of convergence per number of timesteps) by a significant margin in all games. We found that TRPO could only learn two games, Seaquest and Pong, in 10 million timesteps, and performed worse than A2C in terms of sample efficiency.

In Table 1 we present the mean of rewards of the last 100 episodes in training for 50 million timesteps, as well as the number of episodes required to achieve human performance [16] . Notably, on the games Beamrider, Breakout, Pong, and Q-bert, A2C required respectively 2.7, 3.5, 5.3, and 3.0 times more episodes than ACKTR to achieve human performance. In addition, one of the runs by A2C in Space Invaders failed to match human performance, whereas ACKTR achieved 19723 on average, 12 times better than human performance (1652). On the games Breakout, Q-bert and Beamrider, ACKTR achieved 26%, 35%, and 67% larger episode rewards than A2C.

We also evaluated ACKTR on the rest of the Atari games; see Appendix for full results. We compared ACKTR with Q-learning methods, and we found that in 36 out of 44 benchmarks, ACKTR is on par with Q-learning methods in terms of sample efficiency, and consumed a lot less computation time. Remarkably, in the game of Atlantis, ACKTR quickly learned to obtain rewards of 2 million in 1.3 hours (600 episodes), as shown in Figure 2. It took A2C 10 hours (6000 episodes) to reach the same performance level.

## 5.2 Continuous control

We ran experiments on the standard benchmark of continuous control tasks defined in OpenAI Gym [5] simulated in MuJoCo [26], both from low-dimensional state-space representation and directly from pixels. In contrast to Atari, the continuous control tasks are sometimes more challenging due to high-dimensional action spaces and exploration. The results of eight MuJoCo environments trained for 1 million timesteps are shown in Figure 3. Our model significantly outperformed baselines on six out of eight MuJoCo tasks and performed competitively with A2C on the other two tasks (Walker2d and Swimmer).

We further evaluated ACKTR for 30 million timesteps on eight MuJoCo tasks and in Table 2 we present mean rewards of the top 10 consecutive episodes in training, as well as the number of

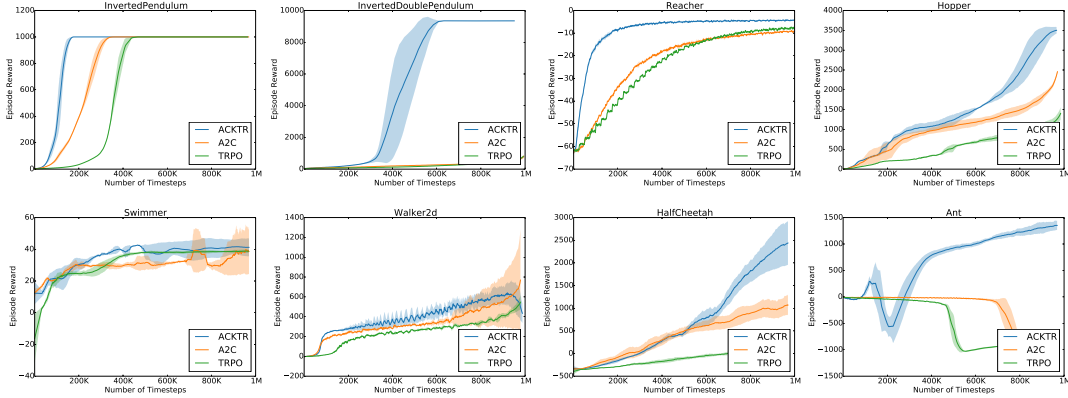

Figure 3: Performance comparisons on eight MuJoCo environments trained for 1 million timesteps (1 timestep equals 4 frames). The shaded region denotes the standard deviation over 3 random seeds.

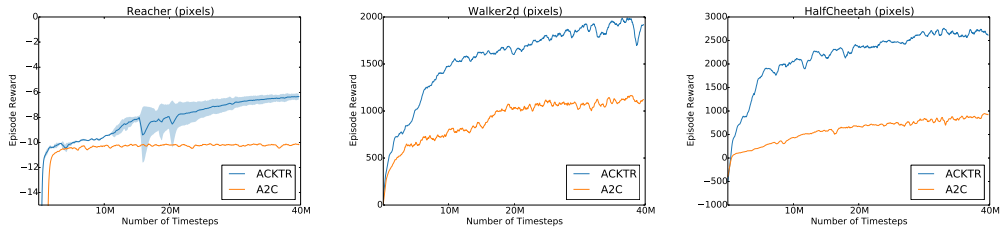

Figure 4: Performance comparisons on 3 MuJoCo environments from image observations trained for 40 million timesteps (1 timestep equals 4 frames).

episodes to reach a certain threshold defined in [7]. As shown in Table 2, ACKTR reaches the specified threshold faster on all tasks, except for Swimmer where TRPO achieves 4.1 times better sample efficiency. A particularly notable case is Ant, where ACKTR is 16.4 times more sample efficient than TRPO. As for the mean reward score, all three models achieve results comparable with each other with the exception of TRPO, which in the Walker2d environment achieves a 10% better reward score.

We also attempted to learn continuous control policies directly from pixels, without providing low-dimensional state space as an input. Learning continuous control policies from pixels is much more challenging than learning from the state space, partially due to the slower rendering time compared to Atari (0.5 seconds in MuJoCo vs 0.002 seconds in Atari). The state-of-the-art actor-critic method A3C [17] only reported results from pixels on relatively simple tasks, such as Pendulum, Pointmass2D, and Gripper. As shown in Figure 4 we can see that our model significantly outperforms A2C in terms of final episode reward after training for 40 million timesteps. More specifically, on Reacher, HalfCheetah, and Walker2d our model achieved a 1.6, 2.8, and 1.7 times greater final reward compared to A2C. The videos of trained policies from pixels can be found at https://www.youtube.com/watch?v=gtM87w1xGoM. Pretrained model weights are available at https://github.com/emansim/acktr.

### 5.3 A better norm for critic optimization?

The previous natural policy gradient method applied a natural gradient update only to the actor. In our work, we propose also applying a natural gradient update to the critic. The difference lies in the norm with which we choose to perform steepest descent on the critic; that is, the norm $||\cdot||_B$ defined in section 2.2. In this section, we applied ACKTR to the actor, and compared using a first-order method (i.e., Euclidean norm) with using ACKTR (i.e., the norm defined by Gauss-Newton) for critic optimization. Figures 5 (a) and (b) show the results on the continuous control task HalfCheetah and the Atari game Breakout. We observe that regardless of which norm we use to optimize the critic, there are improvements brought by applying ACKTR to the actor compared to the baseline A2C. However, the improvements brought by using the Gauss-Newton norm for optimizing the critic are more substantial in terms of sample efficiency and episode rewards at the end of training. In addition,

|  |  | ACKTR | | A2C | | TRPO | |
|---|---|---|---|---|---|---|---|
| Domain | Threshold | Rewards | Episodes | Rewards | Episodes | Rewards | Episodes |
| Ant | 3500 (6000) | 4621.6 | **3660** | 4870.5 | 106186 | **5095.0** | 60156 |
| HalfCheetah | 4700 (4800) | 5586.3 | **12980** | 5343.7 | 21152 | **5704.7** | 21033 |
| Hopper | 2000 (3800) | **3915.9** | **17033** | 3915.3 | 33481 | 3755.0 | 39426 |
| IP | 950 (950) | 1000.0 | **6831** | 1000.0 | 10982 | 1000.0 | 29267 |
| IDP | 9100 (9100) | 9356.0 | **41996** | **9356.1** | 82694 | 9320.0 | 78519 |
| Reacher | -7 (-3.75) | **-1.5** | **3325** | -1.7 | 20591 | -2.0 | 14940 |
| Swimmer | 90 (360) | 138.0 | 6475 | **140.7** | 11516 | 136.4 | **1571** |
| Walker2d | 3000 (N/A) | 6198.8 | **15043** | 5874.9 | 26828 | **6874.1** | 27720 |

Table 2: ACKTR, A2C, and TRPO results, showing the top 10 average episode rewards attained within 30 million timesteps, averaged over the 3 best performing random seeds out of 8 random seeds. "Episode" denotes the smallest $N$ for which the mean episode reward over the $N^{th}$ to the $(N + 10)^{th}$ game crosses a certain threshold. The thresholds for all environments except for InvertedPendulum (IP) and InvertedDoublePendulum (IDP) were chosen according to Gu et al. [7], and in brackets we show the reward threshold needed to solve the environment according to the OpenAI Gym website [5].

the Gauss-Newton norm also helps stabilize the training, as we observe larger variance in the results over random seeds with the Euclidean norm.

Recall that the Fisher matrix for the critic is constructed using the output distribution of the critic, a Gaussian distribution with variance $\sigma$. In vanilla Gauss-Newton, $\sigma$ is set to 1. We experimented with estimating $\sigma$ using the variance of the Bellman error, which resembles estimating the variance of the noise in regression analysis. We call this method adaptive Gauss-Newton. However, we find adaptive Gauss-Newton doesn't provide any significant improvement over vanilla Gauss-Newton. (See detailed comparisons on the choices of $\sigma$ in Appendix.)

### 5.4   How does ACKTR compare with A2C in wall-clock time?

We compared ACKTR to the baselines A2C and TRPO in terms of wall-clock time. Table 3 shows the average timesteps per second over six Atari games and eight MuJoCo (from state space) environments. The result is obtained with the same experiment setup as previous experiments. Note that in MuJoCo tasks episodes are processed sequentially, whereas in the Atari environment episodes are processed in parallel; hence more frames are processed in Atari environments. From the table we see that ACKTR only increases computing time by at most 25% per timestep, demonstrating its practicality with large optimization benefits.

| (Timesteps/Second) | Atari | | | MuJoCo | | |
|---|---|---|---|---|---|---|
| batch size | 80 | 160 | 640 | 1000 | 2500 | 25000 |
| ACKTR | 712 | 753 | 852 | 519 | 551 | 582 |
| A2C | 1010 | 1038 | 1162 | 624 | 650 | 651 |
| TRPO | 160 | 161 | 177 | 593 | 619 | 637 |

Table 3: Comparison of computational cost. The average timesteps per second over six Atari games and eight MuJoCo tasks during training for each algorithms. ACKTR only increases computing time at most 25% over A2C.

### 5.5   How do ACKTR and A2C perform with different batch sizes?

In a large-scale distributed learning setting, large batch size is used in optimization. Therefore, in such a setting, it is preferable to use a method that can scale well with batch size. In this section, we compare how ACKTR and the baseline A2C perform with respect to different batch sizes. We experimented with batch sizes of 160 and 640. Figure 5 (c) shows the rewards in number of timesteps. We found that ACKTR with a larger batch size performed as well as that with a smaller batch size. However, with a larger batch size, A2C experienced significant degradation in terms of sample efficiency. This corresponds to the observation in Figure 5 (d), where we plotted the training curve in terms of number of updates. We see that the benefit increases substantially when using a larger batch size with ACKTR compared to with A2C. This suggests there is potential for large speed-ups with ACKTR in a distributed setting, where one needs to use large mini-batches; this matches the observation in  [2].

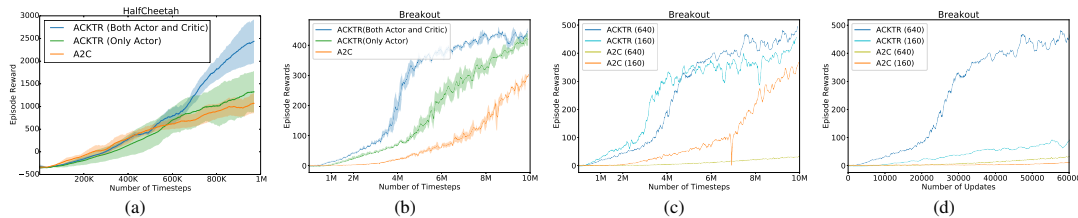

Figure 5: (a) and (b) compare optimizing the critic (value network) with a Gauss-Newton norm (ACKTR) against a Euclidean norm (first order). (c) and (d) compare ACKTR and A2C with different batch sizes.

## 6 Conclusion

In this work we proposed a sample-efficient and computationally inexpensive trust-region-optimization method for deep reinforcement learning. We used a recently proposed technique called K-FAC to approximate the natural gradient update for actor-critic methods, with trust region optimization for stability. To the best of our knowledge, we are the first to propose optimizing both the actor and the critic using natural gradient updates. We tested our method on Atari games as well as the MuJoCo environments, and we observed 2- to 3-fold improvements in sample efficiency on average compared with a first-order gradient method (A2C) and an iterative second-order method (TRPO). Because of the scalability of our algorithm, we are also the first to train several non-trivial tasks in continuous control directly from raw pixel observation space. This suggests that extending Kronecker-factored natural gradient approximations to other algorithms in reinforcement learning is a promising research direction.

### Acknowledgements

We would like to thank the OpenAI team for their generous support in providing baseline results and Atari environment preprocessing codes. We also want to thank John Schulman for helpful discussions.

## Footnotes

[2]The A2C and TRPO Atari baseline results are provided to us by the OpenAI team, `https://github.com/openai/baselines`.

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
