[Supplementary Material · acktr_supplementary.pdf]

# Supplementary Materials for Scalable trust-region method for deep reinforcement learning using Kronecker-factored approximation

**Yuhuai Wu**[*]
University of Toronto
Vector Institute
ywu@cs.toronto.edu

**Elman Mansimov**[*]
New York University
mansimov@cs.nyu.edu

**Shun Liao**
University of Toronto
Vector Institute
sliao3@cs.toronto.edu

**Roger Grosse**
University of Toronto
Vector Institute
rgrosse@cs.toronto.edu

**Jimmy Ba**
University of Toronto
Vector Institute
jimmy@psi.utoronto.ca

## A Experimental details

### A.1 Discrete control

For experiments on the Atari Environment, we adopted the same input preprocessing procedure as in [3], with a slight modification to the architecture. Specifically, we used a shared network to parameterize the policy and value function: The first convolutional layer is of 32 filters of size $8 \times 8$ with stride 4 followed by another convolutional layer with 64 filters of size $4 \times 4$ and stride 2, followed by a final convolutional layer with 32 filters of size $3 \times 3$ with stride 1, followed by a fully connected layer of size 512, followed by one softmax output layer that parameterizes the policy and a linear output layer that predicts the value. We used 32 filters in the third convolutional layer because we found that it saved time for computing the Fisher matrix inverse without any degradation in performance. (One alternative would be to use the doubly factored approximation [1] with all 64 filters.) For the baseline A2C, we used the same architecture as in [3]. For TRPO, because of its high per-iteration expense, we used a smaller architecture, with 2 convolutional layers followed by a fully connected layer with 128 units. The first convolutional layer had 8 filters of size $8 \times 8$ with stride 4, followed by another convolutional layer with 16 filters of size $4 \times 4$ with stride 2.

We tuned the maximum learning rate $\eta_{\max}$ using a grid search over $\{0.7, 0.2, 0.07, 0.02\}$ on the game of Breakout, with the trust region radius $\delta$ set to $0.001$. We used the same hyperparameters for all Atari experiments. Both the baseline (A2C) and our method used a linear schedule for the learning rate over the course of training, and entropy regularization with weight $0.01$. Following [3], the agent is trained on each game using 50 million time steps or 200 million frames. Unless otherwise stated, we used a batch size of 640 for ACKTR, 80 for A2C, and 512 for TRPO. The batch sizes were chosen to achieve better sample efficiency.

### A.2 Continuous control

For experiments with low-dimensional state space as an input we used two separate neural networks with 64 hidden units per layer in a two-layer network. We used Tanh and ELU [2] nonlinearities for the policy network and value network, respectively, for all layers except the output layer, which didn't have any nonlinearity. The log standard deviation of a Gaussian policy was parameterized as a

---

[*]Equal contribution.

bias in a final layer of policy network that didn't depend on input state. For all experiments, we used ACKTR and A2C trained with batch sizes of 2500 and TRPO trained with a batch size of 25000. The batch sizes were chosen to be consistent with the experimental design of the results provided by the OpenAI team.

For experiments using pixels as an input we passed in a $42 \times 42$ RGB image along with the previous frame to a convolutional neural network. The two convolutional layers contained 32 filters of size $3 \times 3$ with stride of 2 followed by a fully connected layer of 256 hidden units. In contrast to our Atari experiments, we found that separating the policy network and value function into two separate networks resulted in better empirical performance in both ACKTR and A2C. We used ReLU nonlinearity for the policy network and ELU [2] nonlinearity for the value function. We also found that it is important to use orthogonal initialization for both networks, otherwise the A2C baseline failed to improve its episode reward. All models were trained with batch size of 8000. We tuned the maximum learning rate $\eta_{\max}$ using a grid search over $\{0.3, 0.03, 0.003\}$ on the tasks of Reacher and Hopper, with the trust region radius $\delta$ set to 0.001. We fixed hyperparameters for all MuJoCo experiments.

## B    Results on the remaining Atari games

In this section we present results on the rest of the Atari games in Table 1. The score reported for our method is the mean of the last 100 episode rewards after 50 million time steps. Each episode is started with 30 no-op actions. We find that there is no result reported in A3C [4] or A2C using the same metric. Hence we compare our results with other Q-learning methods obtained from [5]. Due to limited computational resources, we were only able to evaluate ACKTR on a subset of the games. Our results are obtained with a single random seed and we have not tuned any hyperparameters. Although we use only one random seed, our results are on par with Q-learning methods, which use off-policy techniques such as experience replay. Q-learning methods usually take days to finish one training, whereas our method takes only 16 hours on a modern GPU.

Table 1: Raw scores across all games, starting with 30 no-op actions. Other scores from [5].

| GAMES | HUMAN | DQN | DDQN | DUEL | PRIOR. | PRIOR. DUEL. | ACKTR |
|---|---|---|---|---|---|---|---|
| Alien | 7,127.7 | 1,620.0 | 3,747.7 | 4,461.4 | 4,203.8 | 3,941.0 | 3197.1 |
| Amidar | 1,719.5 | 978.0 | 1,793.3 | 2,354.5 | 1,838.9 | 2,296.8 | 1059.4 |
| Assault | 742.0 | 4,280.4 | 5,393.2 | 4,621.0 | 7,672.1 | 11,477.0 | 10,777.7 |
| Asterix | 8,503.3 | 4,359.0 | 17,356.5 | 28,188.0 | 31,527.0 | 375,080.0 | 31,583.0 |
| Asteroids | 47,388.7 | 1,364.5 | 734.7 | 2,837.7 | 2,654.3 | 1,192.7 | 34,171.6 |
| Atlantis | 29,028.1 | 279,987.0 | 106,056.0 | 382,572.0 | 357,324.0 | 395,762.0 | 3,433,182.0 |
| Bank Heist | 753.1 | 455.0 | 1,030.6 | 1,611.9 | 1,054.6 | 1,503.1 | 1,289.7 |
| Battle Zone | 37,187.5 | 29,900.0 | 31,700.0 | 37,150.0 | 31,530.0 | 35,520.0 | 8910.0 |
| Beamrider | 16,926.5 | 8,627.5 | 13,772.8 | 12,164.0 | 23,384.2 | 30,276.5 | 13,581.4 |
| Berzerk | 2,630.4 | 585.6 | 1,225.4 | 1,472.6 | 1,305.6 | 3,409.0 | 927.2 |
| Bowling | 160.7 | 50.4 | 68.1 | 65.5 | 47.9 | 46.7 | 24.3 |
| Boxing | 12.1 | 88.0 | 91.6 | 99.4 | 95.6 | 98.9 | 1.45 |
| Breakout | 30.5 | 385.5 | 418.5 | 345.3 | 373.9 | 366.0 | 735.7 |
| Centipede | 12,017.0 | 4,657.7 | 5,409.4 | 7,561.4 | 4,463.2 | 7,687.5 | 7,125.28 |
| Crazy Climber | 35,829.4 | 110,763.0 | 117,282.0 | 143,570.0 | 141,161.0 | 162,224.0 | 150,444.0 |
| Demon Attack | 1,971.0 | 12,149.4 | 58,044.2 | 60,813.3 | 71,846.4 | 72,878.6 | 274,176.7 |
| Double Dunk | -16.4 | -6.6 | -5.5 | 0.1 | 18.5 | -12.5 | -0.54 |
| Enduro | 860.5 | 729.0 | 1,211.8 | 2,258.2 | 2,093.0 | 2,306.4 | 0.0 |
| Fishing Derby | -38.7 | -4.9 | 15.5 | 46.4 | 39.5 | 41.3 | 33.73 |
| Freeway | 29.6 | 30.8 | 33.3 | 0.0 | 33.7 | 33.0 | 0.0 |
| Gopher | 2,412.5 | 8,777.4 | 14,840.8 | 15,718.4 | 32,487.2 | 104,368.2 | 47,730.8 |
| Ice Hockey | 0.9 | -1.9 | -2.7 | 0.5 | 1.3 | -0.4 | -4.2 |
| James Bond | 302.8 | 768.5 | 1,358.0 | 1,312.5 | 5,148.0 | 812.0 | 490.0 |
| Kangaroo | 3,035.0 | 7,259.0 | 12,992.0 | 14,854.0 | 16,200.0 | 1,792.0 | 3,150.0 |
| Krull | 2,665.5 | 8,422.3 | 7,920.5 | 11,451.9 | 9,728.0 | 10,374.4 | 9,686.9 |
| Kung-Fu Master | 22,736.3 | 26,059.0 | 29,710.0 | 34,294.0 | 39,581.0 | 48,375.0 | 34,954.0 |
| Phoenix | 7,242.6 | 8,485.2 | 12,252.5 | 23,092.2 | 18,992.7 | 70,324.3 | 133,433.7 |
| Pitfall! | 6,463.7 | -286.1 | -29.9 | 0.0 | -356.5 | 0.0 | -1.1 |
| Pong | 14.6 | 19.5 | 20.9 | 21.0 | 20.6 | 20.9 | 20.9 |
| Q-bert | 13,455.0 | 13,117.3 | 15,088.5 | 19,220.3 | 16,256.5 | 18,760.3 | 23,151.5 |
| River Raid | 17,118.0 | 7,377.6 | 14,884.5 | 21,162.6 | 14,522.3 | 20,607.6 | 17,762.8 |
| Road Runner | 7,845.0 | 39,544.0 | 44,127.0 | 69,524.0 | 57,608.0 | 62,151.0 | 53,446.0 |
| Robotank | 11.9 | 63.9 | 65.1 | 65.3 | 62.6 | 27.5 | 16.5 |
| Seaquest | 42,054.7 | 5,860.6 | 16,452.7 | 50,254.2 | 26,357.8 | 931.6 | 1,776.0 |
| Solaris | 12,326.7 | 3,482.8 | 3,067.8 | 2,250.8 | 4,309.0 | 133.4 | 2,368.6 |
| Space Invaders | 1,668.7 | 1,692.3 | 2,525.5 | 6,427.3 | 2,865.8 | 15,311.5 | 19,723.0 |
| Star Gunner | 10,250.0 | 54,282.0 | 60,142.0 | 89,238.0 | 63,302.0 | 125,117.0 | 82,920.0 |
| Time Pilot | 5,229.2 | 4,870.0 | 8,339.0 | 11,666.0 | 9,197.0 | 7,553.0 | 22,286.0 |
| Tutankham | 167.6 | 68.1 | 218.4 | 211.4 | 204.6 | 245.9 | 314.3 |
| Up and Down | 11,693.2 | 9,989.9 | 22,972.2 | 44,939.6 | 16,154.1 | 33,879.1 | 436,665.8 |
| Video Pinball | 17,667.9 | 196,760.4 | 309,941.9 | 98,209.5 | 282,007.3 | 479,197.0 | 100,496.6 |
| Wizard Of Wor | 4,756.5 | 2,704.0 | 7,492.0 | 7,855.0 | 4,802.0 | 12,352.0 | 702.0 |
| Yars' Revenge | 54,576.9 | 18,098.9 | 11,712.6 | 49,622.1 | 11,357.0 | 69,618.1 | 125,169.0 |
| Zaxxon | 9,173.3 | 5,363.0 | 10,163.0 | 12,944.0 | 10,469.0 | 13,886.0 | 17,448.0 |

# C   MuJoCo results with comparisons to OpenAI baselines

We compared the performance results of ACKTR with the results of A2C and TRPO sent to us by the OpenAI team (https://github.com/openai/baselines). We followed their experimental protocol as closely as possible. Like our baselines, A2C and TRPO were trained with the same two-layer architecture with 64 hidden units in each layer on batch sizes of 2500 and 25000 respectively. However, in contrast to our baselines, the value function used Tanh nonlinearities and was "softly" updated by calculating the weighted average of the value function before and after the update.

Compared to our implementation of the A2C baseline, A2C implemented by OpenAI performed better on the Hopper, InvertedPendulum, Swimmer, and Walker2d tasks while performing worse on the Reacher and HalfCheetah tasks. TRPO by OpenAI performed worse than the TRPO trained by us on Hopper while achieving the same performance on the rest of the tasks. Results are shown in Figure 1.

Figure 1: Performance comparisons on seven MuJoCo environments trained for 1 million timesteps (1 timestep equals 4 frames). The numbers for A2C and TRPO were provided to us by the OpenAI team. The shaded region denotes the standard deviation over 3 random seeds.

## D   Adaptive Gauss-Newton?

In this section we investigate whether training the critic using adaptive Gauss-Newton (i.e., keeping an estimate of the standard deviation of the Bellman error as the standard deviation of the critic output distribution) provides any improvement over vanilla Gauss-Newton (both defined in Section 5.3). We ran adaptive Gauss-Newton on all six standard Atari games and eight MuJoCo tasks. The results are shown in Figure 2 and Figure 3. We see that in Atari games, adaptive Gauss-Newton hurts the performance in terms of sample efficiency in Beamrider, Q-bert and Seaquest, and shows only a slight improvement in the game of Pong. In MuJoCo tasks, adaptive Gauss-Newton gives a slight improvement on the tasks of InvertedDoublePendulum, Swimmer, Walker2d, and Ant while performing on par on the tasks of InvertedPendulum and Reacher, and working considerably worse on the HalfCheetah task compared to vanilla Gauss-Newton.

Figure 2: Performance comparisons of critic trained with adaptive Gauss-Newton and vanilla Gauss-Newton on six Atari environments trained for 10 million timesteps (1 timestep equals 4 frames). The shaded region denotes the standard deviation over 2 random seeds.

Figure 3: Performance comparisons of critic trained with adaptive Gauss-Newton and vanilla Gauss-Newton on eight MuJoCo environments trained for 1 million timesteps (1 timestep equals 4 frames). The shaded region denotes the standard deviation over 3 random seeds.

# E    How well does the Kronecker-factored quadratic approximation match the exact KL?

We indirectly tested how accurate the Kronecker-factored approximation to the curvature is by measuring the exact KL changes during training, while performing trust region optimization using a Kronecker-factored quadratic model. We tested this in two Mujoco environments, HalfCheetah and Reacher. The values of approximated KL and exact KL are shown in Figure 4. From the plot we see that exact KL is close to the trust region radius, showing the effectiveness of trust region optimization via Kronecker-factored approximation.

Figure 4: The plot shows the exact KL changes during training with trust region optimization using ACKTR. The actual KL is close to the trust region radius, showing the effectiveness of trust region optimization via Kronecker-factored approximation.