[Reviews · NeurIPS 2017]

Reviewer 1



I believe the primary contribution of paper is to apply K-FAC to RL environment successively with impressive results compared to competing alternatives. This is more an application paper of an existing approach in a new context then a theoretical contribution paper, though they do demonstrate how to adapt K-FAC in this actor/critic context which may not be clear to a general user. Minor details: Line 7: "the Mujoco environment" Line 14: "complex behavior skills" Line 20: "executing" -> permitting Line 35: conjugate gradient -> the conjugate gradient method Line 40: "the curvature information" Line 47: uses "a" Kronecker-factor approximation to "the" natural policy gradient "method" Line 50-53: Our algorithm asynchronously computes second-order Kronecker matrix statistics and approximate inverses with a computational cost that is only ... Line 58: Should argument a depend on t? a -> a_t Line 83. And in practice approximation are used Line 88. "We" denote the input Line 108. Dependencies on tau for p is not present in definition. Line 131. "the" natural gradient "method" is Line 237. to baselines A2C and TRPOP

Reviewer 2



This paper investigates a method to approximating the policy function using a kronecker-factored trust region method. This is applied to deep networks that take as input the image of the game, and uses gradient methods to minimize the discounted cumulative reward. The kronecker factorization appears at each layer, and basically both simplifies the model and makes training time much faster. There is very little new theory in this paper; most follows a previous work [15], which is applied to pure deep learning; in contrast, this paper extends to deep reinforcement learning for games with images as state representations. The main contributions of this paper is in 1) describing how this setup can be applied to deep reinforcement learning, and 2) providing several interesting numerical results on real games. In this aspect I believe the paper is solid; all experiments show clear benefit over existing methods, and from my own experience in reinforcement learning, this problem is well-motivated. The only comment I have is that there is only one work that the author compares against: A2C. Is this really the only other plausible method in this area that attempts to accelerate learning? Overall I believe the paper is well-written, the theory at least well-reviewed (if not extended) and the contribution in the experiments is good. Add'l comments: - lines 220-228: What is meant by Gauss-Newton metric?

Reviewer 3



The manuscript discusses an important topic, which is optimization in deep reinforcement learning. The authors extend the use of Kronecker-Factored approximation to develop a second order optimization method for deep reinforcement learning. The optimization method use kronecker-factored approximation to the Fisher matrix to estimate the curvature of the cost, resulting in a scalable approximation to natural gradients. The authors demonstrate the power of the method (termed ACKTR) in terms of the performance of agents in Atari and Mujoco RL environments, and compare the proposed algorithm to two previous methods (A2C and TRPO). Overall the manuscript is well-written and to my knowledge the methodology is a novel application to Kronecker-factored approximation. I have the following comments to improve the manuscript: 1- I believe the complexity of the ACKTR, A2C and TRPO algorithms are not the same. The authors compare the performance of these algorithms per iteration in most figures, which shows that ACKTR is superior to the other methods. I suggest that the authors make the same comparison with x-axis representing optimization time instead of number of iterations, thus taking into account the complexity of the different algorithms. This is done in Figure 2 for one of the games. I am proposing here that this time comparison also be performed in the other experiments. 2- I wonder how would a method like ADAM perform relative to ACKTR. 3- The details of the algorithm are distributed at different parts of section 3. I think it is important to include a summary of the algorithm at the end of section 3 or at least in the suppmats to make it clearer.